# Stability of Graphene Oxide Composite Membranes in an Aqueous Environment from a Molecular Point of View

Chiara Muzzi [1], Anastasios Gotzias [2], Enrica Fontananova [1] and Elena Tocci [1,*]

[1] Institute on Membrane Technology (ITM–CNR), c/o University of Calabria, Via P. BUCCI, Cubo 17C, 87036 Rende, Italy; c.muzzi@itm.cnr.it (C.M.); e.fontananova@itm.cnr.it (E.F.)

[2] National Centre for Scientific Research "Demokritos", Institute of Nanoscience and Nanotechnology INN, 15310 Athens, Greece; a.gotzias@inn.demokritos.gr

* Correspondence: e.tocci@itm.cnr.it or elena.tocci@cnr.it; Tel.: +39-0984-402038; Fax: +39-0984-402103

**Abstract:** We used molecular dynamics to investigate the stability of graphene oxide (GO) layers supported on three polymeric materials, namely a polyvinylidene fluoride (PVDF), a pristine and a crosslinked polyamide–imide (PAI and PAI-cr). The membrane configurations consisted of a few layers of GO nanosheets stacked over the specified polymeric supports and submerged in water. We monitored the position, the tilt angle, and the radial distribution function of the individual GO nanosheets in respect to the plane of the supports. We showed that the outermost GO nanosheets were more distorted than those attached directly on the supports. The greatest distortion was observed for the GO nanosheets of the PVDF-supported system. Next, we recorded the density profiles of the water molecules across the distance from the layers to the polymer and discussed the hydrogen bonds between water hydrogens and the oxygen atoms of the GO functional groups.

**Keywords:** molecular simulation; computational chemistry; GO; supported GO polymeric systems; stability

## 1. Introduction

Carbon-based membranes are considered promising alternatives to membrane materials currently applied in water treatment and nanofiltration technology [1–4]. This is due to the unique properties of a miscellany of carbon nanoparticles and carbonaceous frameworks, which exhibit different textures and nanosized pore networks, having large surface areas, excellent thermal and mechanical strengths, competitive antibacterial performances, and adequate adsorption/selectivity capacities. Specifically, graphene oxides (GOs) are two-dimensional (2D) layered materials, with oxygen-containing functional groups (epoxy, hydroxyl, and carboxyl) grafted on the basal plane and the periphery of graphene nanosheets. The GO sheets pile up in multilayer formations via π-complexation interactions with tunable interlayer distances [5,6]. The π complexes involve the noncovalent bonding of the sp2 carbons of the adjoining GO sheets. Multilayer formations of GOs can operate like molecular sieves [7–10], likely letting only the water molecules pass through their nanosized spacings, while excluding the metal ions from an aqueous solution. In this regard, GO materials are widely applied in seawater desalination and purification [5,7–11], energy storage and conversion [12], solvent dehydration, and gas separation [13].

Free-standing graphene oxides suffer from poor mechanical strength and swelling defects in polar and aqueous environments [14,15]. The reason is that the oxygen groups make the graphitic surface hydrophilic. Water molecules may access the inner core of the GO matrix and be adsorbed by the functional groups between the layers. Water loading may limit the strength of the π interactions of the adjoining nanosheets and, locally, increase the interlayer distance. This may cause the GO layers to detach from the initial columnar structures producing few-layer GO assemblies [15,16].

The tendency of GOs to delaminate in aqueous solvents reduces the applicability of GO membranes in water-related processes. Several methods have been suggested to overcome this hurdle, such as chemical crosslinking [17–19], molecular bridging [20], physical confinement using polymer matrix [6], cationic bonding and intercalation [7,21], and the formation of a graphene–polyelectrolyte multilayer [22].

Recently, our group produced a water-stable GO membrane [23]. The sample was prepared using solvent evaporation of dispersed GO deposited on a crosslinked co-poly(amide-imide) film (PAI-cr) [24]. Different supports were also tried for comparison. In this respect, hydrophilic supports were used such as polyether sulfones (PES), cellulose acetate (CA), co-poly(amide-imide) Torlon® (PAI), and its crosslinked derivative (PAI-cr). Hydrophobic polymeric films were also used, such as polyvinylidene fluoride (PVDF), polypropylene (PP), and polytetrafluoroethylene (PTFE). It was demonstrated that the hydrophobic supported systems were unstable due to the unfavored interactions between the films and the dispersed GOs. For instance, Figure 1a shows that the GO-PVDF sample appeared fragmented and crushed on contact with water, whereas other systems having a hydrophilic support such as GO-PAI (Figure 1b) and GO-PAI-cr (Figure 1c) appeared rather compact and uniform. We noticed some cracks on the edge of the GO-PAI sample, whereas the rest of the structure appeared condensed. Evidently, the crosslinked GO-PAI-cr membrane remained stable after prolonged contact time with water (>30 days). Figure 1d,e show the cross section of a membrane in which the GO was deposited on a PVDF support. The GO layer immediately detached from the support on contact with water (the red arrow indicates the GO layer detached form the support). On the contrary, when the GO was deposited under similar conditions on a PAI-cr membrane, good adhesion between the GO and the support was observed thanks to the electrostatic interaction between the two oppositely charged materials. Figure 1f shows the cross section of GO-PAI-cr membrane after more than 30 days in water.

Complementary to the experimental characterization and analysis of the pristine and the crosslinked polyamide–imide GO composites (GO-PAI and GO-PAI-cr) and the GO-PVDF, it is useful to investigate the binding affinity of the specified hybrid membrane structures at the molecular level using computer simulation.

In the literature, many theoretical studies have shown the transport and selectivity of water and salt solutions inside GO layered membranes [25–31]. The growth in computational power has now made it possible to simulate large GO and polymeric structures with characteristics closer to the real material, providing novel insights into the structural and transport behavior (water flux) relationship [32–36].

However, the phenomenon of the stability of GO composite membranes has not yet been visualized; it remains a challenge for which theoretical approaches can provide a useful insight. This is also due to the lack of possibility of a direct comparison between laboratory data and the models: the laboratory systems are made of a thousand of layers of GO, whereas, for the sake of computational efficiency, model configurations are constrained to be comprised of few layers with finite area sizes. Nevertheless, association/dissociation effects of layered structures are mainly balanced by the interfacial properties and the short-range interactions of the solvents and the support, rather than the actual thickness of their multi-walled configurations. Typically, in MD simulations, periodic boundary conditions are applied, meaning that the simulation box is surrounded by exact replicas of itself in each spatial dimension. In this regard, we may focus on the interfacial structure of GO layers having an infinite area size, by simply applying the periodicity in the lateral dimensions and by designing the simulation box accordingly.

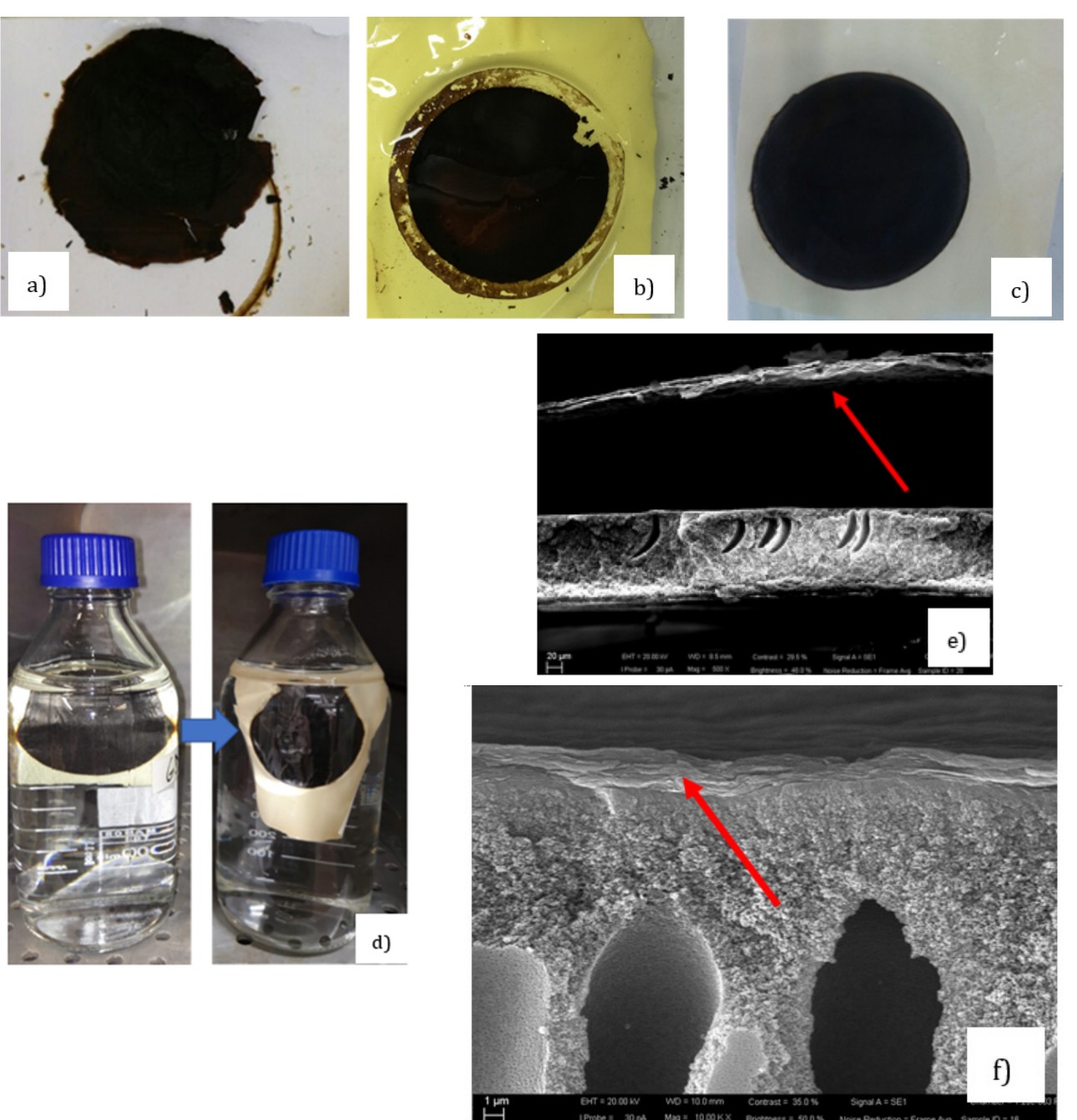

**Figure 1.** Picture of (**a**) GO-PVDF, (**b**) GO-PAI, (**c**) GO–PAI-cr immersed in water at room temperature immediately after their preparation; (**d**) GO-PAI-cr immersed in water at 60 °C for 30 days (left: as immersed; right after 30 days); (**e**) SEM image of the GO layer detached from the PVDF support and (**f**) SEM image of the GO layer adhering well on PAI-cr (red arrow shows the GO layer position).

In this work, MD simulations were used to highlight the different stability of supported GO systems on three polymeric materials: Polyvinylidene fluoride, the pristine Polyamide-imide PAI, and the crosslinked Polyamide-imide PAI-cr. The behavior of supported GO systems was compared with graphene oxide layers in water. The results follow similar performances of experimental results and explain the nature of the strong interaction observed.

## 2. Modelling

We performed the simulations using the BIOVA package [37]. We used the Condensed-phase Optimized Molecular Potentials for Atomistic Simulation Studies (COMPASS) force-field [38]. This forcefield is specifically developed to simulate the dynamic and structure properties of polymers. We used both van der Waals and electrostatic interactions and set default atomic charges on the systems based on the atom definitions [38,39]. We employed the cubic spline truncation method. The cut-off radius was 1.5 Å, and the spline and the buffer widths were 1 Å and 0.5 Å, respectively. In the NPT molecular dynamics simulations (constant number of particles, pressure, and temperature), we used the Nose Hoover Thermostat [40] and the Berendsen barostat [41] to regulate the temperature and the pressure. The thermostat Q ratio was 0.01. We set the barostat decay constant to 0.1 ps. The timestep of the molecular dynamics simulations was 1 fs. We collected data every $10^4$ timesteps.

### 2.1. Graphene Oxide Model

To build the GO monolayer, we used an oxygen content that matched that of the experimental samples [23,24]. Generally, the protonation or deprotonation of the hydroxyl groups and carboxyl groups (acid groups) is a function of the pH [42]. In our model, we assumed that the groups were in the neutral state (-OH and -COOH). The weight percentage of the monolayer was 59.5% C, 39.2% O, and 1.3% H, which was close to the elemental analysis of the actual GO samples used in the experiments (49–56% C, 41–50% O, 0–1% H) [42]. We introduced hydroxyl and epoxy groups on the basal plane of a graphene sheet and carboxyl groups on its periphery. The functional groups were distributed randomly on both sides of the sheet. Specifically, we used 87 epoxy groups, 101 hydroxyl groups, and 35 carboxyl groups. The resulting GO monolayer contained 916 atoms, of which 522 atoms were carbons (C), 258 oxygens (O), and 136 hydrogens (H).

The GO monolayer was optimized using steepest gradient energy minimization. Next, we built a three-layer structure of GOs by placing 12 GO monolayers, as illustrated in Figure 2. We set the interlayer distance at 9 Å. This was the initial configuration of a bidimensional 50% superposition of the GO nanosheets. We used full periodicity in the simulation box. The box dimensions were $115 \times 76 \times 32.21$ Å$^3$.

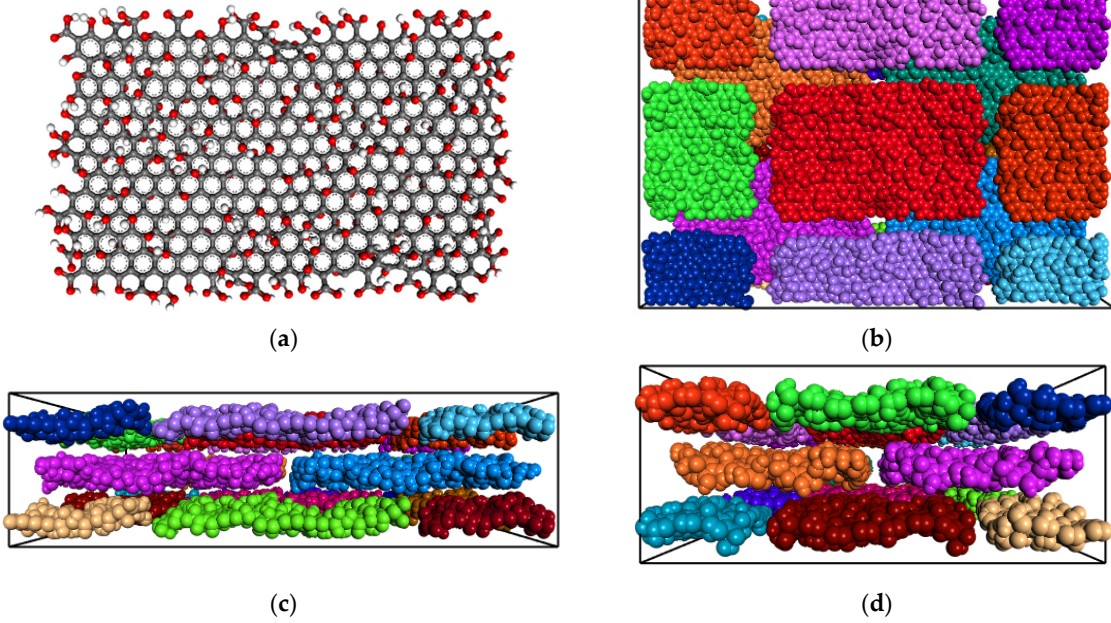

(a)  (b)

(c)  (d)

**Figure 2.** (**a**) GO layer structure. (**b**) Top view of GO box containing 3 layers of GO, each comprising 4 sheets. (**c**) Long and (**d**) short side view of the GO layers. In panels (**b**–**d**), each sheet is colored differently to make it easy to understand their spatial organization. Atoms are pictured with respect to their van der Waals radii.

The multilayer GO structure was solvated in water by setting the water density to $1 \text{ g/cm}^3$. The solvation process introduced 15,936 water molecules (47,808 atoms) into the box. We performed a steepest gradient minimization followed by an NPT simulation over 0.5 ns to ensure that the water molecules were not overlapping with the Connolly isosurface (i.e., the surface traced by the van der Waals radii of the atoms) of the GO nanosheets. Consequently, we performed a long NPT simulation over 4 ns, using a timestep of 1 fs for data production.

### 2.2. Polymeric Models

We considered three polymeric materials to be used as supports to deposit the GO layers, namely a polyvinylidene fluoride (PVDF), a pristine and a crosslinked polyamide–imide (PAI and PAI-cr). The polymeric supports are presented in Table 1.

**Table 1.** Chemical structures of supporting polymers (numbers refer to the initial boxes, not the supercells).

| Name | Abbr. | Chemical Structure | Monom. in a Chain | # Atoms per Chain | Chain in a Layer |
|---|---|---|---|---|---|
| Polyvinylidene fluoride | PVDF | | 300 | 1802 | 5 |
| Polyamide-imide (PAI) | PAI | | 50 | 1787 | 2 |
| crosslinked Polyamide-imide | PAI-cr | | 25 | 900 | 4 |

We built a long chain of n monomers of each material ($n_{PVDF} = 300$, $n_{PAI} = 50$, $n_{PAI-cr} = 25$) by branching the monomers to each other with a random torsion [43–45]. The chain conformations of the macromolecules were sampled in 1000 steps. We set the macromolecules to fill a confined amorphous layer using a density of $2.3 \text{ g/cm}^3$, at 300 K. In practice, we used a simulation box with one side that was one-third of the desired box size to speed up the sampling of polymer conformations. The dimensions of the actual simulation box were obtained using a super cell of the smaller box. We used the amorphous cell algorithm [37,46] to grow the macromolecule by inserting one segment at each Monte Carlo step. [47]. The trial conformations of the inserted segments were sampled according to Flory's RIS theory [48]. We used argon as a spacer to promote uniform growth in the cell and prevent segment accumulations in local high-density areas. The PVDF film contained 5 chains and 900 argon atoms. The PAI films contained 2 chains and 300 argon atoms. The argon atoms were deleted prior the NPT equilibration of the polymeric chains. The polymeric films were also put between two layers of graphene to prevent the chains fluctuating over the plane of the membrane. The positions of the graphenes were restrained during the equilibration. We performed Forcite annealing dynamics in five cycles. The initial temperature was 300 K, and the mid-cycle temperature was 600 K. We used five heating ramps per cycle with 1000 NVT (constant number of particles, volume, and temperature) steps each. The resulting configuration was further equilibrated using an NPT simulation over 3 ns at 298 K and 1 bar. At the end of the equilibration the graphene layers were deleted, and the

box sizes were adjusted to exclude the resulting empty space. The final layer dimensions were ($112.73 \times 74.66 \times 41.91$) Å$^3$ for PVDF and ($114.02 \times 75.35 \times 39.20$) Å$^3$ for PAI.

The crosslinked PAI was created starting from 25-monomer-long 180 torsion chains of PAI and DAMP molecules as linkers. PAI chains were modified to a semi-reacted state where the imide groups were opened; the monomers were linked by a single amide group with a simple carbon atom in its ortho position. In addition, DAMP molecules were modified so that they carry the amide group at both their ends. Four PAI chains and 50 DAMPs were inserted into a confined amorphous layer normal to the z axis at a density of 2 g cm$^{-3}$ together with 300 Argon atoms as spacers. Graphene walls were inserted on the top and on the bottom of the layer to prevent chain exiting from z-perpendicular faces. A custom-made code was used to perform the actual crosslink of the chains.

The crosslinking code detected the available reactive atoms and analyzed their mutual positions. The closest couple of atoms was analyzed. If their distance was below a set threshold, a second check regarding the possible bond-angle was performed. When the possible bond-angle values lay within a pre-set range of the actual value, a bond was formed, and the geometry was optimized over $10^4$ steps of conjugate gradient minimization. If one or both geometrical checks failed, a short dynamic was performed (NVT ensemble dynamics, 298 K, 1 ps). After five cycles without any new bond, annealing was performed (between 300 K and 350 K, with five cycles of NVT ensemble dynamics of 10 ps). Both bond-length and bond-angle range were increased gradually until the formation of new bonds was feasible. A crosslinking degree of 74.9% was achieved. All unreacted DAMP molecules were deleted from the system. When the polymerization progress ended, the imidic groups between the monomers where no crosslinking occurred were not restored. We aimed to highlight the differences between the two chemical groups (imidic and amidic) on the affinity between GO and PAI/PAI-cr. Crosslinked PAI required a longer equilibration than the pristine PAI. Some geometrical equilibrations were performed together with some short dynamics (10–100 ps STP) and some increasingly hotter annealing (minimum temperature 280 K, maximum temperature 320 K and 400 K). Once the polymeric layer was equilibrated, the graphene walls were deleted.

### 2.3. Multilayer Models

The supported GO models were prepared as a multilayer, containing, from bottom to top, a Graphene wall, a polymeric substrate, the three-layer GO system, a volume of water (at density 1 g cm$^{-3}$ of dimensions $115 \times 76 \times 32.21$ Å), and a second Graphene wall. The distance between the lowest GO sheets and the polymeric substrate was about 2 Å for PAI, about 4 Å for PVDF, and about 4 Å for crosslinked PAI. The small changes in the distances depended on local roughness of the polymeric layer that prevented the GO from approaching the support too closely without overlapping the branching groups of the polymer. Graphene walls and GO sheets were constrained during an equilibration run over 1 ns NVT at 298 K, then the constraints were removed. The amount of water present in every system is reported in Table 2. Each multilayer geometry was optimized and then NPT dynamics was performed for 4 ns at 298 K and 1 bar for data production. The dimensions of the system before data production dynamics are also reported in Table 2.

**Table 2.** Multilayer supported systems details.

| System | Water Content (n° mol.) | Dimensions (Å$^3$) | Total Number of Atoms |
|--------|--------|--------|--------|
| GO-PAI | 18,576 | $113.0 \times 76.1 \times 114.9$ | 105,579 |
| GO-PVDF | 21,720 | $113.0 \times 76.0 \times 114.8$ | 109,878 |
| GO-PAI-cr | 20,868 | $115.1 \times 76.2 \times 124.0$ | 121,902 |

## 3. Results and Discussion

Figure 3 shows the configurations of free-standing and supported GO layers on PVDF, PAI, and PAI-cr films. Within water, the layers appeared highly dispersed due to the oxygen-containing functionalities. The water content on the surface enhanced the repulsive nature of the interlayer interactions, making the GO nanosheets shift their rotation angle.

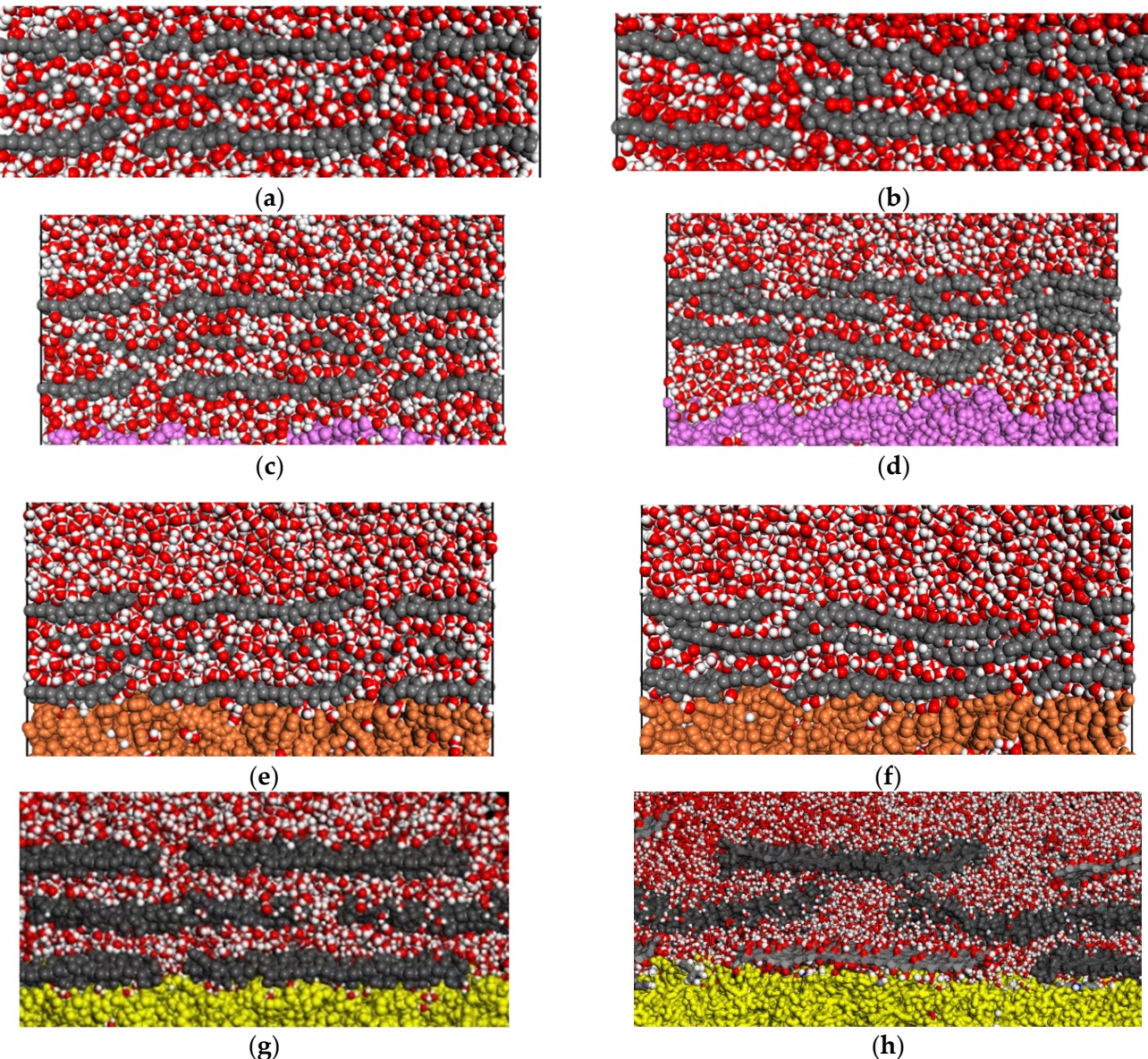

(a)

(b)

(c)

(d)

(e)

(f)

(g)

(h)

**Figure 3.** GO sheets at the beginning and end of the simulation: (**a**,**b**) GO in water; (**c**,**d**) PVDF (pink); (**e**,**f**) PAI (orange); (**g**,**h**) PAI–cr (yellow). In all pictures, the colors are O = red; H = white, GO layer = deep grey.

In the case of the supported systems the inner layers were more anchored to the polymeric films. Several water molecules were encountered between the polymer and the nanosheets of the first layer. The immobilization of the nanosheets should be attributed to non-covalent bonds of the water molecules that bridge the polymeric surface and the GO. It has been observed in the literature that these interactions are associated with strong hydrogen bonds formed between the water hydrogens and the oxygens of the functional groups of the nanosheets [49,50]. Representative hydrogen bonds of water

molecules bridging the polymer and the first GO layer are shown in Figure 4, as snapshots corresponding to the initial and final configurations of the NPT equilibrations.

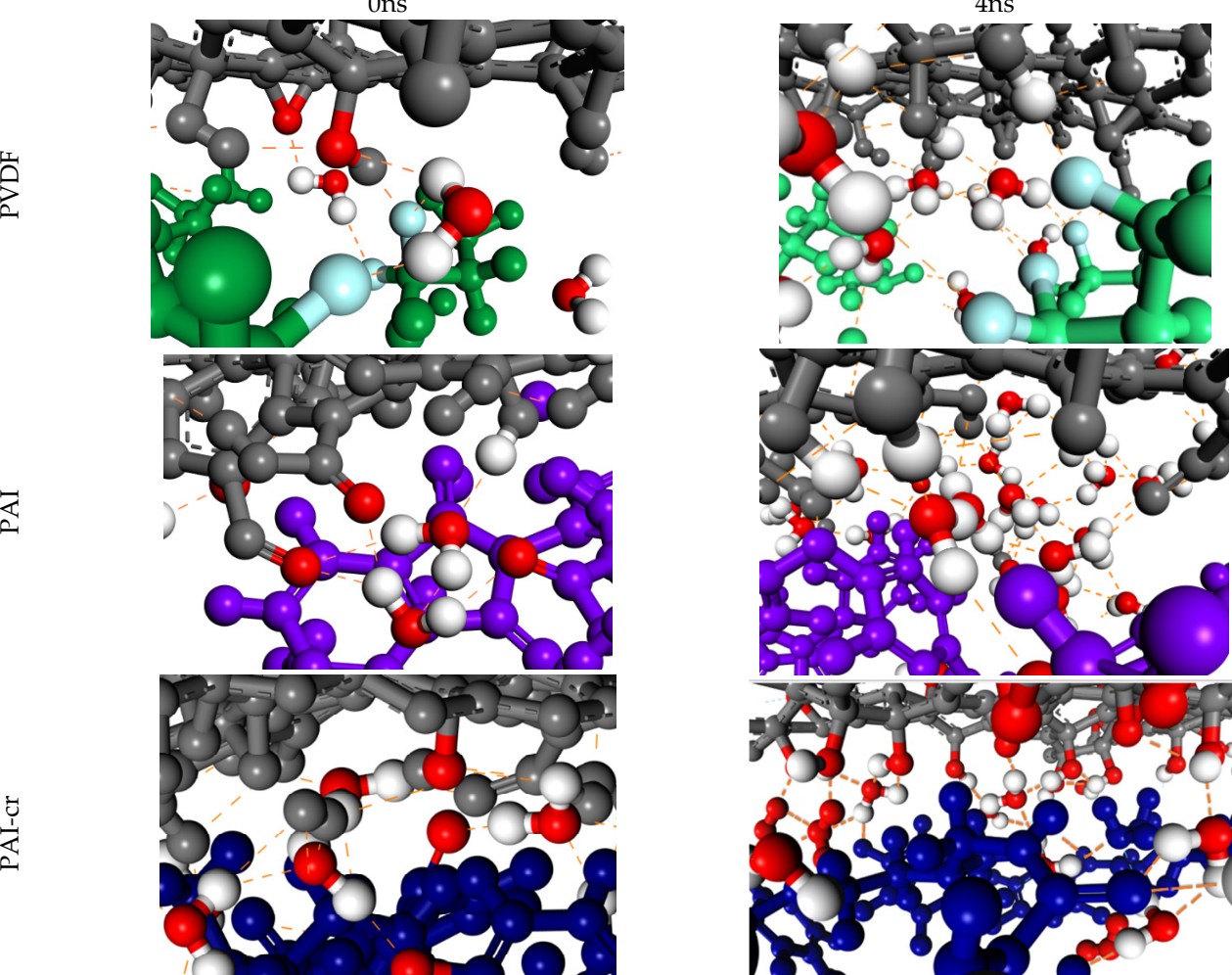

**Figure 4.** Examples of hydrogen bonds between polymer–water–GO in the lower layer of GO at the beginning of the simulation (**left column**) and at the end (**right column**). GO in colored in gray, PAI in purple, PVDF in green, and crosslinked PAI in blue. Some atom types are colored differently in order to highlight their atom type: oxygen in red, hydrogen in white, fluorine in light blue. The orange dotted line represents the hydrogen bonds.

Figure 5 shows the density profiles of water molecules as a function of the distance from the top edge of the simulation box and through the graphene oxide layers. The profile curves depict the water densities in the initial and the final configurations of the NPT simulations. The boundary of the stacks with the graphene oxide layers is highlighted using two vertical yellow dashed lines. Both the supported and the unsupported systems display highly ordered structures of water molecules in the interlayer space at the initial configurations. These structures reflect the parallel orientation of the graphene nanosheets. The density plateau on the left side of the plots corresponds to the bulk density of water. On the right side of the plots, we encounter a negligible density. This density corresponds to a small amount of water molecules inserted inside the polymeric layer upon solvation.

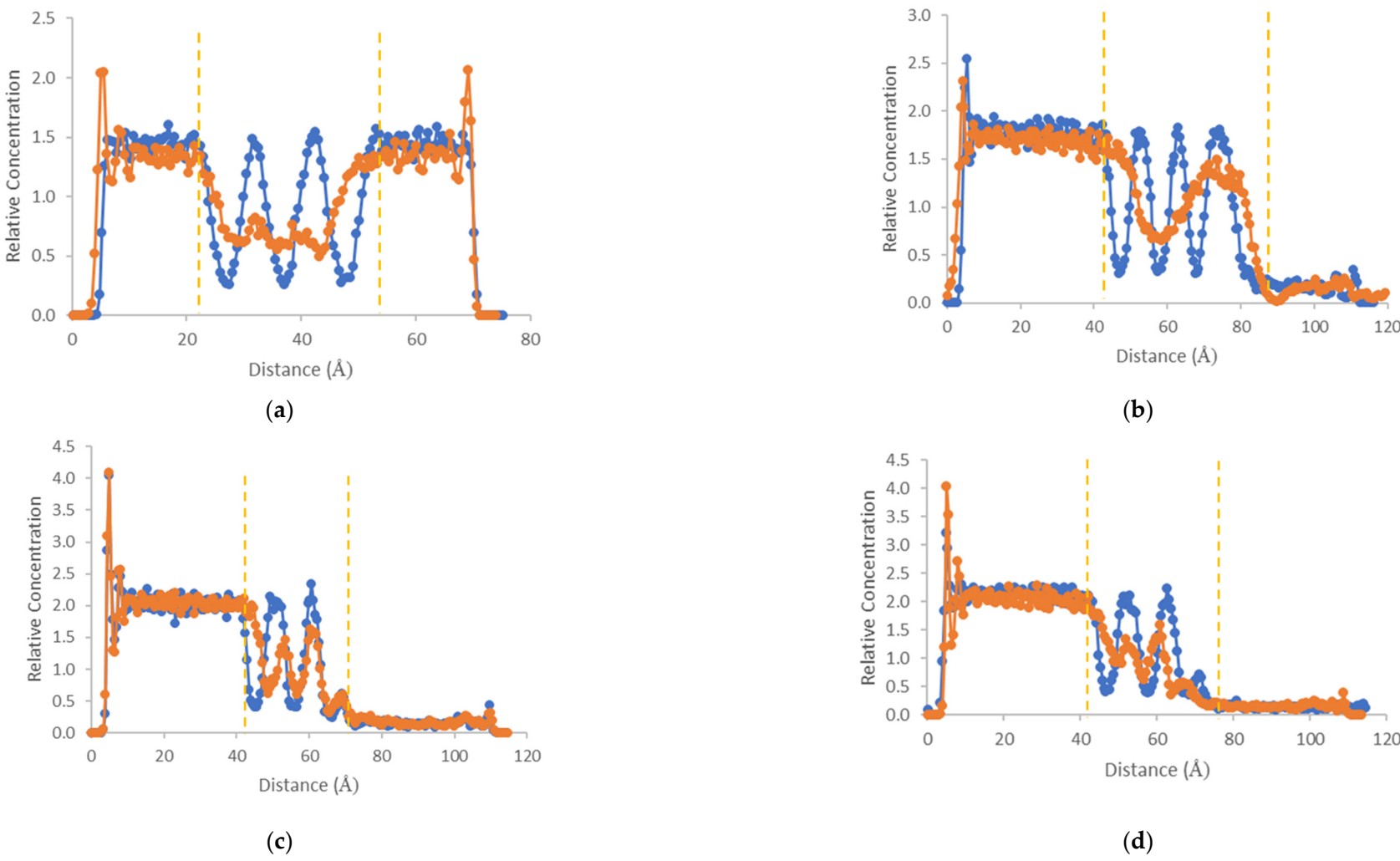

**Figure 5.** Density profile perpendicular to GO sheets at the beginning of the simulation (blue lines) and at the end (orange lines) for (**a**) GO in water; (**b**) PVDF; (**c**) PAI; and (**d**) PAI-cr, as a function of the distance from the top edge of the box. The yellow dashed lines denote the boundaries of the stacks of the GO layers.

On the other hand, the water density profiles at the final configurations show that the water structure has been disturbed due to oscillations of the GO layers after a simulation over 4 ns. The interlayer distance is similar to that of the initial configuration. However, this value corresponds to the center-of-mass distance of the GO layers. The nanosheets on the layers were not oriented exactly parallel. Nevertheless, using a hydrophilic support, the initial ordering of the GO layers was partially maintained throughout the equilibration. The situation was even better for the GO-PAI-cr system. This is also depicted by the shape of the corresponding water density profiles in the interlayer range of distances.

Figure 6 shows the position and the orientation angle of the individual GO nanosheets of the different membrane systems. The GO nanosheets are indexed from 1 to 12, where the labels GO1 up to GO4 correspond to the nanosheets of the outermost layer, the GO5 to GO8 nanosheets lie at the middle layer, and the GO9 to GO12 nanosheets are attached to the polymeric film. We will now discuss the initial and final configurations of the specified nanosheets. The differences of the center-of-mass displacement of the nanosheets between the layers are marginal. We may note that the outermost nanosheets are displaced more than those in direct contact with the support. This is especially true for the PVDF-supported membrane, in which the polymeric film is hydrophobic.

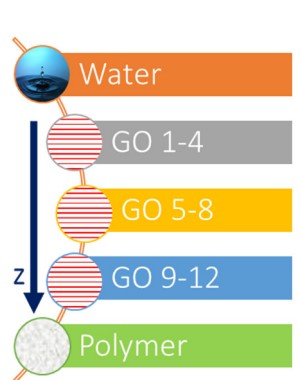

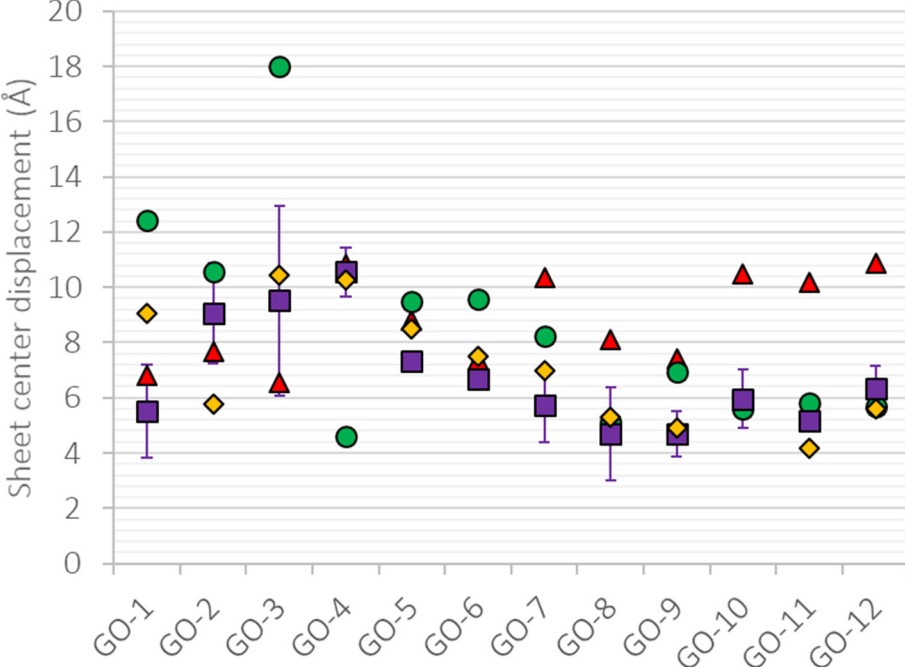

**Figure 6.** GO sheet three-dimensional displacement: GO in water (red triangles), GO-PAI (purple squares), GO-PVDF (green circles), and GO-PAI-cr (yellow rhombuses).

In Figure 7, we present the tilt angle of the individual nanosheets in respect to the plane of the polymeric film. The tilt angle decreases for the layers closer to the support. This confirms the partial stabilization effect of the polymeric substrate. The nanosheets of the PVDF supported membrane obtain the greatest rotation. Both PAI-GO membranes are shown to be efficiently stable where we observe small GO rotations. In the case of the pristine PAI-GO, the rotation angle of the nanosheets is more uniform throughout the layers.

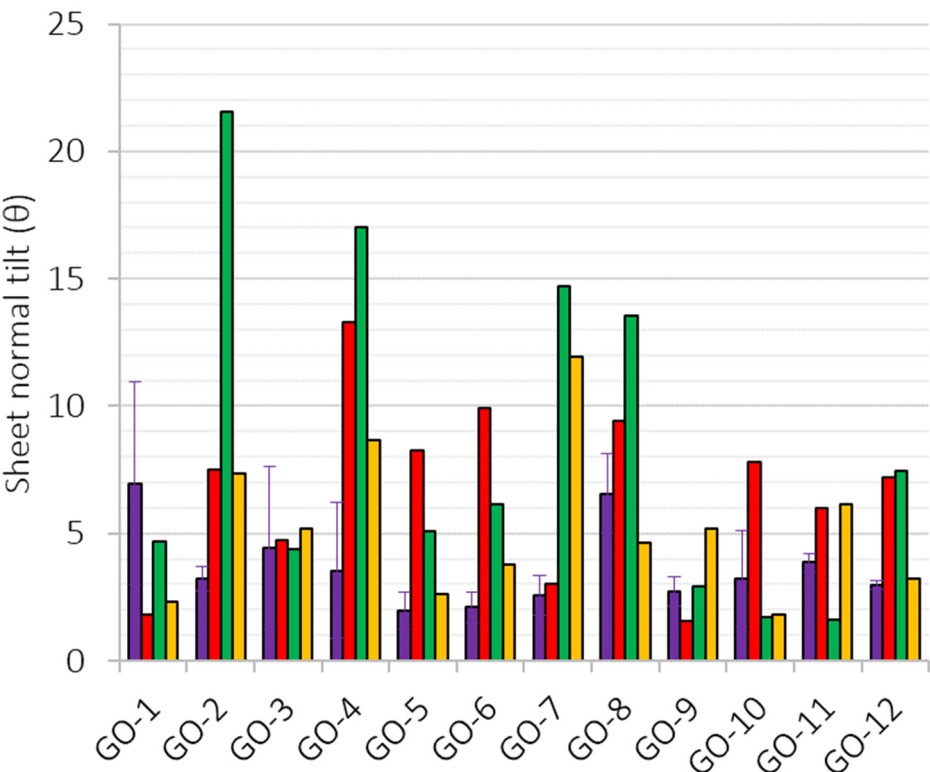

**Figure 7.** GO sheet tilt angle with respect to normal axes: GO in water (red), GO-PAI (purple), GO-PVDF (green), and GO-PAI-cr (yellow). We investigated the stability in water of GOs supported on different polymeric films using molecular dynamics simulations. The membranes consisted of three layers of GOs deposited onto the hydrophobic and hydrophilic polymeric supports. We showed that the GOs of the outermost layer were more distorted than those of the first. We observed the greatest distortion for the GO layers of the PVDF support due to the hydrophobic characteristics of the substrate. We also observed that the water density profile curves flatten compared to the initial configurations, because the solvent phase is rearranged upon equilibration due to the reorientation of the GO layers. The reorientation of the layers is detected by monitoring the tilt angle of the nanosheets in respect to the plane of the support. We visualized that water hydrogens bridge the polar branches of the polymeric film with the oxygen groups of the adjacent GO nanosheets.

## 4. Conclusions

Molecular dynamics simulations were used to investigate the different stability of graphene oxide (GO) supported on three polymers: Polyvinylidene fluoride, pristine Polyamide-imide, and crosslinked Polyamide-imide. This analysis was compared with the behavior of graphene oxide layers in water. We noted that the outermost GO nanosheets were more distorted than those nanosheets in direct contact with the polymeric support. This was confirmed by recording the displacement and the rotational angle of the GO nanosheets and by observing the water density profiles along the distance from the plane of the support and through the GO layers. We have discussed the configurations observed at the initial and final timestep of the MD simulations. We have shown that the PVD supported systems are less stable compared to the PAI and the PAI-cr supported systems.

**Author Contributions:** Conceptualization, C.M., A.G., E.F. and E.T.; methodology, C.M., A.G. and E.T.; visualization, E.T.; writing, C.M. and E.T. All authors have read and agreed to the published version of the manuscript.

**Funding:** The Italian Ministry of Education University and Research (prot. MIUR no. 10912, 06/06/2016, concession grant decree no. 3366, 18 December 2018, IDEA-ERANETMED2-72-357 is gratefully acknowledged.

**Data Availability Statement:** Not applicable.

**Conflicts of Interest:** The authors declare no conflict of interest.

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
