# Peer review of "Stability of Graphene Oxide Composite Membranes in an Aqueous Environment from a Molecular Point of View"

_applsci, doi:10.3390/app12073460_

Round 1

Reviewer 1 Report

The study "Stability of graphene oxide (GO) composite membranes in an aqueous environment: a molecular view" considers the graphene oxide system stabilized by polymer chains (PVDF, PAI)  as a perspective membrane filters. Nevertheless the study is theoretical, it has experimental basis, which increases it's value. The molecular dynamics protocol is carefully planned and explained in details in the text. The results of study are scientifically sound and could be qualitatively applicable for a broader class of complexes (2d sheets + polymer chains).

However, the clarity of the presentation must be improved. The suggestions are:

  1. remove abbreviation GO from the title
  2. don't use extra spaces around dash, in a words like "non-covalent", "few-layer", etc.
  3. Figure 1 is spread on two pages. Strongly suggest to render this and other figures as a single image, with labels (a, b, etc) embedded in it.
  4. page 4. "We collected data every 10e4 timesteps." 10e4 is confusing. Is it 10 000 or 100 000? Don't use engineering E-notation.
  5. Does the selection of "87 epoxy groups, 101 hydroxyl groups and 35 carboxyl groups" correspond to any specific pH of solution?
  6. last paragraph on the page 4. Should there be reference to Figure 2, not Figure 1?
  7. same paragraph. Use upper index instead of ^ in "Å^3."
  8. Caption of Figure 2. The explanation "(CPK scale 1)" is unnecessary. Instead, state that van der Walls (not wan-) radii are for panels b,c,d, not for panel a.
  9. Table 1 is placed too far from it's first mention
  10. page 6. What is "10000-geometry optimization" ?
  11. Chemical structure of PVDF in Table 1 is not readable
  12. Unify the notation of Cr-PAI in Table 1 with other text (PAI-Cr). Also, I would suggest to use lower-case "cr", not to confuse with chromium element.
  13. Table 2. Put Angstrom units in column caption, check the last column caption: should it be "Total number of atoms"?
  14. Figure 3 - figure caption is totally absent
  15. page 9 "left side of the plots corresponds to the bulk
    density of water.". The left side value is zero. Does it mean that relative density of bulk water is zero? Correct figures or that statement, if not.
  16. Figure 5 - remove horizontal grid lines, decrease marker size,
    explain orange dash lines

Author Response

Reviewer #1: The study "Stability of graphene oxide (GO) composite membranes in an aqueous environment: a molecular view" considers the graphene oxide system stabilized by polymer chains (PVDF, PAI) as a perspective membrane filters. Nevertheless the study is theoretical, it has experimental basis, which increases it's value. The molecular dynamics protocol is carefully planned and explained in details in the text. The results of study are scientifically sound and could be qualitatively applicable for a broader class of complexes (2d sheets + polymer chains). 

Answer: We appreciate the reviewer for his comments and suggestions. We have revised the manuscript according to the comments. Our answers to their notes are listed below. 

However, the clarity of the presentation must be improved. The suggestions are:

  1. remove abbreviation GO from the title

Answer: The abbreviation is deleted.

  1. don't use extra spaces around dash, in words like "non-covalent", "few-layer", etc.

Answer: The extra spaces are removed

  1. Figure 1 is spread on two pages. Strongly suggest to render this and other figures as a single image, with labels (a, b, etc.) embedded in it.

Answer: We have merged the figure panels in one figure. And made the modification suggested by the reviewer.

  1. page 4. "We collected data every 10e4 timesteps." 10e4 is confusing. Is it 10 000 or 100 000? Don't use engineering E-notation.

Answer: This is 10000. We now use superscripts instead of scientific notations when labeling large numbers.

  1. Does the selection of "87 epoxy groups, 101 hydroxyl groups and 35 carboxyl groups" correspond to any specific pH of solution?

Answer: To build the GO monolayer, we used an oxygen content that matched that of the experimental samples. [23,24] Generally, the protonation or deprotonation of the hydroxyl groups and carboxyl groups (acid groups) is a function of the pH. [42] In our model we have assumed that the groups are in the neutral state (-OH and -COOH).

[23] Fontananova, E.; Tocci, E.; Abu-Zurayk, R.; Grosso, V.; Meringolo, V.; Muzzi, C.; Di Profio, G. An environmental-friendly electrostatically driven method for preparing graphene oxide composite membranes with amazing stability in aqueous solutions. submitted to J. Membr. Sci.

[24] Graphenea, S. Graphene_datasheet.

[42] Ederer, J.; Janoš, P.; Ecorchard, P.; Štengl, V.; Bělčická, Z.; Št’astný, M.; Pop-Georgievski, O.; Dohnal, V. Quantitative determination of acidic groups in functionalized graphene by direct titration. Reactive and Functional Polymers 2016, 103, 44–53, doi:10.1016/j.reactfunctpolym.2016.03.021.

  1. last paragraph on the page 4. Should there be reference to Figure 2, not Figure 1?

Answer: The reviewer is correct. This is a reference for Figure 2.

  1. same paragraph. Use upper index instead of ^ in "Å^3."

Answer: We now use a superscript.

  1. Caption of Figure 2. The explanation "(CPK scale 1)" is unnecessary. Instead, state that van der Walls (not wan-) radii are for panels b,c,d, not for panel a.

Answer: We corrected the caption according to the comment.

  1. Table 1 is placed too far from its first mention

Answer: Table 1 is now placed at a different location, near its first mention.

  1. page 6. What is "10000-geometry optimization"?

Answer: This a conjugate gradient energy minimization over 104 steps

  1. Chemical structure of PVDF in Table 1 is not readable

Answer: we changed the scheme modifying the structure of PVDF.

  1. Unify the notation of Cr-PAI in Table 1 with other text (PAI-Cr). Also, I would suggest to use lower-case "cr", not to confuse with chromium element.

Answer: We now use only the PAI-cr notation.

  1. Table 2. Put Angstrom units in column caption, check the last column caption: should it be "Total number of atoms"?

Answer: The reviewer is correct. We have inserted Angstrom units in the column caption.

  1. Figure 3 - figure caption is totally absent

Answer: We thank the reviewer for the comment. We guess the caption must have been removed upon compiling the pdf from the source. 

The caption of Figure 3 is:

GO sheets at the beginning of the simulation and at the end of: a) and b) GO in water; c) and d) PVDF (pink); e) and f) PAI (orange); g) and h) PAI–cr (yellow). In all pictures the colors are O = red; H= white, GO layer = deep grey.

  1. page 9 "left side of the plots corresponds to the bulk
    density of water.". The left side value is zero. Does it mean that relative density of bulk water is zero? Correct figures or that statement, if not.

Answer: The reviewer is correct. The densities are expressed as a function of the distance from the top edge of the box. The density at high distances corresponds to the water density within the polymeric film, which, as expected, is negligible.

  1. Figure 5 - remove horizontal grid lines, decrease marker size,
    explain orange dash lines

Answer: The orange lines denote the boundaries of the stacks of the GO layers. We removed the minor grid lines on the y axis of the plots.

Reviewer 2 Report

The topic of the article is interesting. The authors used simulation to explain the stability of graphene oxide supported on three polymeric materials.

1.- Correct the chemical structure of Polyvinylidene fluoride of the table 1.

2.- Check figure 3, description not found.

3. -Some papers determine the permeability of water in GO membranes, have you considered it?

Author Response

Reviewer #2: The topic of the article is interesting. The authors used simulation to explain the stability of graphene oxide supported on three polymeric materials.

 Answer: We appreciate the reviewer, for his comments. We have revised the manuscript according to his suggestions and notes.

1.- Correct the chemical structure of Polyvinylidene fluoride of the table 1.

Answer: we changed the scheme modifying the structure of PVDF.

2.- Check figure 3, description not found.

Answer: We thank the reviewer for the comment. We guess the caption must have been removed upon compiling the pdf from the source. 

The caption of Figure 3 is:

GO sheets at the beginning of the simulation and at the end of: a) and b) GO in water; c) and d) PVDF (pink); e) and f) PAI (orange); g) and h) PAI–cr (yellow). In all pictures the colors are O = red; H= white, GO layer = deep grey.

  1. -Some papers determine the permeability of water in GO membranes, have you considered it?

Answer: We study the interfacial properties of GO layers on three polymeric supports. We do this using molecular dynamics. At first place, we establish the methodology required to design these systems so that we can perform the relevant simulations. Modeling the diffusivity of solvent molecules through the GOs necessitates a different type of simulation protocol. In brief, it requires to steer a single (or more) solvent molecule(s) on a path crossing the pore network shaped by the walls of the GOs and collect the energy distributions along the coordinates of this path. This is the subject of our next study.  

Reviewer 3 Report

  1. The caption of figure 3 is missing.
  2. Figure 1 e and f are not described in the text. Some descriptions must be added.
  3. The molecular structure of PVDF in Table 1 is not correctly presented.
  4. Since the MD simulation in this work also contains finite number of atoms, how can it be interpreted to experimental results as the authors claimed in Introduction?
  5. A summary of all simulations data and analysis is necessary before the Conclusion section.

Author Response

Reviewer #3:

  1. The caption of figure 3 is missing.

 Answer: Answer: We thank the reviewer for the comment. We guess the caption must have been removed upon compiling the pdf from the source. 

The caption of Figure 3 is:

GO sheets at the beginning of the simulation and at the end of: a) and b) GO in water; c) and d) PVDF (pink); e) and f) PAI (orange); g) and h) PAI–cr (yellow). In all pictures the colors are O = red; H= white, GO layer = deep grey.

  1. Figure 1 e and f are not described in the text. Some descriptions must be added.

 Answer: We now give details for these figures in the revised text

  1. The molecular structure of PVDF in Table 1 is not correctly presented.

 Answer: we changed the scheme modifying the structure of PVDF.

  1. Since the MD simulation in this work also contains finite number of atoms, how can it be interpreted to experimental results as the authors claimed in Introduction?

Answer: Typically, in MD simulations, periodic boundary conditions are applied, meaning that the simulation box is surrounded by exact replicas of itself in each spatial dimension. Here, we employ periodicity only on the lateral dimensions, meaning that our samples have an infinitely large surface. This setup is required to study the interfacial properties of the membrane layers, of the support and the water–walls interactions. On the other hand, properties on the interface which contribute to the stability of the membrane depend mainly on short-range interactions (i.e., ranging no more than the distance of three layers) rather than the actual thickness of the membrane. In this regard our modeling systems provide a fair description of the associated laboratory samples.

  1. A summary of all simulations data and analysis is necessary before the Conclusion section.

Answer: This is the summary of the results:

We investigated the stability in water of GOs supported on different polymeric films using molecular dynamics simulations. The membranes consisted of three layers of GOs deposited onto the hydrophobic and hydrophilic polymeric supports. We showed that the GOs of the outermost layer were more distorted than those of the first. We observed the greatest distortion for the GO layers of the PVDF support, due to the hydrophobic characteristics of the substrate. We also observed that the water density profile curves flatten compared to the initial configurations, because the solvent phase is rearranged upon equilibration due to the reorientation of the GO layers. The reorientation of the layers is detected by monitoring the tilt angle of the nanosheets in respect to the plane of the support. We visualized that water hydrogens bridge the polar branches of the polymeric film with the oxygen groups of the adjacent GO nanosheets.
